

# Comparing trophic position estimates using bulk and compound specific stable isotope analyses: applying new approaches to mackerel icefish *Champsocephalus gunnari*

Jose Antonio Canseco[1], Edwin J. Niklitschek[2], Claudio Quezada-Romegialli[3], Chris Yarnes[4] and Chris Harrod[5,6]

[1] Centro Oceanográfico de Cádiz, Instituto Español de Oceanografía, Cádiz, Cádiz, Spain
[2] Centro i mar, Universidad de Los Lagos, Chile, Puerto Montt, Los Lagos, Chile
[3] Plataforma de Monitoreo Genómico y Ambiental, Departamento de Química, Facultad de Ciencias, Universidad de Tarapacá, Tarapaca, Tarapaca, Chile
[4] Stable Isotope Facility, UC Davis, Davis, CA, United States of America
[5] Instituto de Ciencias Naturales Alexander Von Humboldt, Universidad de Antofagasta, Antofagasta, Antofagasta, Chile
[6] Nucleo Milenio Invasal, Concepcion, Biobio, Chile

Corresponding authors
Jose Antonio Canseco,
jantonio.canseco@ieo.csic.es
Edwin J. Niklitschek,
edwin.niklitschek@ulagos.cl

## ABSTRACT

Quantifying the tropic position (TP) of an animal species is key to understanding its ecosystem function. While both bulk and compound-specific analyses of stable isotopes are widely used for this purpose, few studies have assessed the consistency between and within such approaches. *Champsocephalus gunnari* is a specialist teleost that predates almost exclusively on Antarctic krill *Euphausia superba*. This well-known and nearly constant trophic relationship makes *C. gunnari* particularly suitable for assessing consistency between TP methods under field conditions. In the present work, we produced and compared TP estimates for *C. gunnari* and its main prey using a standard bulk and two amino acid-specific stable isotope approaches (CSI-AA). One based on the difference between glutamate and phenylalanine ($TP_{Glx-Phe}$), and the other on the proline-phenylalanine difference ($TP_{Pro-Phe}$). To do that, samples from *C. gunnari*, *E. superba* and four other pelagic invertebrate and fish species, all potential prey for *C. gunnari*, were collected off the South Orkney Islands between January and March 2019, analyzed using standard isotopic ratio mass spectrometry methods and interpreted following a Bayesian approach. Median estimates ($CI_{95\%}$) for *C. gunnari* were similar between $TP_{bulk}$ (3.6; $CI_{95\%}$: 3.0-4.8) and $TP_{Glx-Phe}$ (3.4; $CI_{95\%}$:3.2-3.6), and lower for $TP_{Pro-Phe}$ (3.1; $CI_{95\%}$:3.0-3.3). TP differences between *C. gunnari* and *E. superba* were 1.4, 1.1 and 1.2, all compatible with expectations from the monospecific diet of this predator ($\Delta TP=1$). While these results suggest greater accuracy for Glx-Phe and Pro-Phe, differences observed between both CSI-AA approaches suggests these methods may require further validation before becoming a standard tool for trophic ecology.

# INTRODUCTION

Understanding trophic relationships between species presents a significant challenge, reflecting both the problems in characterizing consumer diets (*Harrod & Stallings, 2022*) as well as those related to the large number of potential interactions and the intricate structures that characterize many food-webs (*DeNiro & Epstein, 1981*; *Hussey et al., 2014*; *Quezada-Romegialli et al., 2018*). Various techniques, such as stomach content analysis, bulk stable isotope analysis, and fatty acid analysis, have been employed to quantify trophic interactions and establish links between primary producers and consumers (reviewed in *Harrod & Stallings, 2022*). Although bulk stable isotope analysis has increasingly become the method of choice in recent years (*Pethybridge et al., 2018*), several limitations have been described when using $\delta^{15}N_{bulk}$, such as the potential non-linearity of the nitrogen trophic enrichment factor (TEF) at upper trophic levels (*Hussey et al., 2014*), the sensitivity of this factor to prey type (*Caut, Angulo & Courchamp, 2009*), prey baseline and temperature effects (*Canseco, Niklitschek & Harrod, 2022*; *Canseco, Niklitschek & Harrod, 2022*) and the difficulty in obtaining appropriate isotopic baselines (*Post, 2002*; *Needoba et al., 2003*).

Compound-specific stable isotopic analysis (CSI-AA) is a promising alternative or complement to more conventional bulk SIA (B-SIA) where the analysis targets specific amino acids and/or fatty acids, particularly those that cannot be synthesized by consumers and, thus, are little to no affected by metabolic processes (*Ishikawa, 2018*). Processes that may affect amino acids are fractionation during assimilation and catabolic processes, *e.g.*, if the essential component is part in an biochemical pathway the remaining pool might get enriched compared to its source (*McClelland & Montoya, 2002*). This technique allows for a much more direct tracking of carbon and nitrogen sources along the trophic chain (*Macko et al., 1987*; *Nielsen, Popp & Winder, 2015*; *Ohkouchi et al., 2017*). However, it is more expensive and slower than bulk analysis, leading to relevant trade-offs between resolution and sample size.

To determine trophic position through $\delta^{15}N$, researchers have traditionally applied a concept known as the trophic enrichment factor (TEF or $\Delta$), which represents the combined effect of absorption, fractionation, routing and other less known processes leading to a well-known enrichment of $^{13}C$ and $^{15}N$ in consumers in respect of their prey (*DeNiro & Epstein, 1976*). It is important to note that both CSI-AA and B-SIA require prior estimations or assumptions of TEF values. In B-SIA, the TEF is commonly assumed to be known (∼3‰ for $\delta^{15}N$) and constant (*DeNiro & Epstein, 1978*; *DeNiro & Epstein, 1981*; *Barton et al., 2019*) although it has been observed to have large variability, ranging from 0.0 to 5.2‰ (*Minagawa & Wada, 1984*; *McCutchan et al., 2003*; *Caut, Angulo & Courchamp, 2009*; *Canseco, Niklitschek & Harrod, 2022*). The rationale is somewhat different in the case of CSI-AA, where researchers discriminate between "trophic" and "source" amino acids (*Whiteman et al., 2019*). Similar to the case in bulk samples, isotopic values in trophic

amino acids are influenced by $^{13}$C and $^{15}$N enriching processes, such as transamination and deamination, whose effects accumulate throughout the food web (TEF>0). In contrast, isotopic values in source amino acids remain unaffected by metabolism (*i.e.,* TEF =0) and close to those observable at the base of the food web. These isotopic differences between trophic and source amino acids can be used to estimate trophic position, as long as TEF values become available. These CS-SIA TEF values have been found to be specific to each pair of trophic and source amino acids being compared (*Chikaraishi et al., 2009*; *McMahon & McCarthy, 2016*; *Ramirez et al., 2021*).

The estimation of trophic position using bulk SIA or CSI-AA inherently involves several sources of variation and model assumptions regarding the parameters needed in the estimation method. In the case of bulk SIA, the formula proposed by *Cabana & Rasmussen (1996)* requires researchers to assume a TEF, as mentioned before. However, it has been demonstrated that this assumption is a primary source of variability due to its close relationship with the trophic position of a species which varies across tissues, species and habitats (*Hussey et al., 2014*; *Kjeldgaard, Hewlett & Eubanks, 2021*). Another confounding effect when utilizing bulk SIA to estimate TP is the selection of a baseline organism that serves as the reference trophic position ($\lambda$). This is challenging due to the substantial spatial and temporal variability in the $\delta^{15}$N of potential baseline organisms at the base of the trophic web (*Post, 2002*; *Magozzi et al., 2017*). Regarding the TP estimation using CSI-AA, three main sources of variability have been shown to possibly affect TP estimates. These include, $\beta$, representing the difference in the amino acids phenylalanine (Phe) and glutamic acid (Glx) for marine phytoplankton (*Chikaraishi et al., 2009*), the TEF assumed for trophic and source AA (*Bradley et al., 2015*; *McMahon & McCarthy, 2016*; *Ohkouchi et al., 2017*) and the actual trophic and source AA applied in the TP calculaton (*Larsen et al., 2013*; *Nielsen, Popp & Winder, 2015*; *McMahon & McCarthy, 2016*).

Antarctic krill *Euphausia superba* is a crucial species in Antarctic waters, serving as the primary food source for several predators and linking lower and epipelagic trophic levels with upper and mesopelagic trophic levels (*Trathan & Hill, 2016*; *Trathan et al., 2021*). The mackerel icefish *Champsocephalus gunnari* is a semipelagic fish whose diet composition is almost dominated by *E. superba* with secondary prey, such as *Antarctomysis maxima, Themisto gaudichaudii, Euphausia frigida,* and *Electrona antarctica,* also contributing to its diet (*Jones et al., 2009*; *Kock et al., 2012*; *Zhu & Zhu, 2022*). *Champsocephalus gunnari* was over-exploited during the 60s, 70s and 80s until the population collapsed in the early 90s resulting in the closure of the main fishery in Area 48 (*Kock, 1985*; *Kock & Jones, 2005*). Antarctic marine waters are divided into several management areas, according to the Convention for the Conservation of Antarctic Marine Living Resources (CCAMLR). Within such areas, Area 48 includes the Weddell and Scotia seas and is the area where the South Orkney Islands and the Antarctic peninsula are located. Due to the uncertainty associated with the abundance estimates for the South Orkney Islands (Sub-area 48.2) and the Antarctic Peninsula (Sub-area 48.1) the fishery here remains closed in order to allow the species to recover (*Frolkina, 2002*; *Fallon et al., 2016*). Although the trophic ecology of *C. gunnari* has been studied using various techniques, including stomach content analysis (SCA, *Jones et al., 2009*; *Kock et al., 2012*), fatty acid analysis (*Zhu & Zhu, 2022*)

and a complementary SCA+SIA approaches (*Canseco et al., 2023*), there are no studies estimating its trophic position in Antarctic food-webs. Quantifying the trophic position of *C. gunnari* and its associated food web allows us to detect any ecosystem shifts in Antarctic food webs that have the potential of affecting the recovery of this species.

In this study we estimate the trophic position of *C. gunnari,* its main prey, *E. superba* and secondary prey species, off the South Orkney Islands using bulk and compound specific stable isotope analyses, under different TEF assumptions. By comparing these results, we tested the null hypothesis that trophic position estimates are consistent across SIA estimation techniques. Utilizing a Bayesian approach will help us quantify the uncertainty surrounding the TP of *C. gunnari* and its primary prey across various estimation methods. This will enable us to pinpoint areas requiring further research to achieve consistency among estimation techniques.

## MATERIAL AND METHODS

### Sample collection

Sampling took place between January and March 2019 onboard the RV Kronprins Haakon and the FV Antarctic Endeavour in waters off 60°39′S and 45°00′W. Field sampling was approved by the Instituto Antartico Chileno (RT_68-18) and by the Office of the Commissioner from the Falkland Islands Government House (2018/046). For bulk stable isotope analysis, data were collected as previously described in *Canseco et al. (2023)*. Specifically, we sampled muscle tissue from 284 individual *C. gunnari*, 75 pooled samples (combination of 3–5 similarly-sized individuals) of its main prey *E. superba*, muscle tissue from *E. antarctica* ($n = 30$), a secondary fish prey item, and pooled samples from invertebrate prey species *A. maxima*, *T. gaudichaudii*, and *E. frigida* (total $n = 50$). A total of 23 particulate organic matter (POM) samples were collected by filtering seawater (4 l) through pre-combusted 0.7 µm GF/F Whatman filters. A total of 32 samples were analyzed for nitrogen CSI-AA, distributed as follows: five samples per size-class of *C. gunnari* ($n = 10$), five samples per size-class of *E. superba* ($n = 10$), three samples of *T. gaudichaudii*, three samples of *E. frigida*, three samples of *A. maxim* a and three samples of *E. antarctica*. No samples of POM were analyzed for amino acid $\delta^{15}$N.

### Stable isotope analysis

To analyze the stable isotope ratios of invertebrate species and particulate organic matter (POM), we homogenized 3-5 similarly-sized individuals (pooled samples) from the same species and tow, as previously performed by *Canseco et al. (2023)*. In the case of *C. gunnari* and *E. antarctica,* muscle tissue was collected from individual animals. Muscle tissue and invertebrate pooled samples were rinsed with freshwater, oven dried for 48 h (60 °C) and homogenized, 1 mg of homogenized powder was encapsulated in tin capsules (*Jardine et al., 2003*). Stable isotope ratios $\delta^{13}$C and $\delta^{15}$N were analyzed using a Thermo Scientific Delta V mass spectrometer with a dual inlet and Conflo IV interface connected to a Costech 4010 elemental analyzer (EA), and a high-temperature conversion elemental analyzer (TCEA) at the Center for Stable Isotopes of the University of New Mexico (UNM). Replicated measures of reference materials (UNM CSI protein standard #1, UNM CSI protein

standard #4) showed that precision ($\pm$SD) for $\delta^{13}$C and $\delta^{15}$N values were 0.04 and 0.11 ‰ for carbon and nitrogen respectively, as described in *Canseco et al. (2023)*. We followed *Coplen*'s *(2011)* guidelines for stable isotopes reporting, and our measurements are relative to international standards atmospheric-N$_2$ for $\delta^{15}$N and Vienna PeeDee Belemnite for $\delta^{13}$C.

## Compound specific stable isotope analysis

Amino acids were made suitable for gas chromatography-combustion-isotope ratio mass spectrometry (GC-C-IRMS) by derivatization as N-acetyl methyl esters (NACME; *Corr, Berstan & Evershed, 2007*; *Yarnes & Herszage, 2017*). Prior to derivatization, amino acids were liberated from sample material proteins by acid hydrolysis. GC-C-IRMS was performed on a Thermo Trace GC 1310 gas chromatograph coupled to a Thermo Scientific Delta V Advantage isotope-ratio mass spectrometer *via* a GC IsoLink II combustion interface. Each sample was analyzed twice, but, if the initial measurements showed a very large measurement error, more replicates were analyzed. Replicates of the reference materials for the quality and assurance were measured every five samples, as performed by *Athaillah, Yarnes & Wang (2023)*. The individual amino acids of high purity utilized in quality assurance mixtures were individually calibrated by elemental analysis-isotope ratio mass spectrometry (EA-IRMS) using the primary isotopic reference material for each element (*i.e.,* VPDB for $\delta^{13}$C and air for $\delta^{15}$N).

EA-IRMS was performed using secondary reference materials calibrated against certified standard reference materials from the US Geological Service (USGS), the National Institute of Standard and Technology (NIST), and the International Atomic Energy Agency (IAEA) (*i.e.,* IAEA-600, USGS40, USGS41, USGS42, USGS43, USGS61, USGS64, and USGS65). Calibration procedures for CSIA were performed as previously described in *Athaillah, Yarnes & Wang (2023)*. They were applied identically across reference and sample materials, and initially, the isotopic values for all amino acids were adapted in order to obtain the already known isotopic composition of an internal reference material (*e.g.,* Nor). Afterwards, the stable isotopes of individual amino acids were adjusted using the initial reference mixture for quality assurance, UCD AA1, as a basis, in order to obtain the already known isotopic composition of each amino acid within the mixture. Finally, all measurements were standardized to the primary reference materials for $\delta^{15}$N (*i.e.,* Air) using the second quality assurance reference mixture, UCD AA2. The final evaluation of quality was founded on the precision and unbiasedness of the control materials, which contained a calibrated amino acid mixture, UCD AA3, as well as multiple natural substances.

For summarizing nitrogen SIA results, amino acids were divided into "trophic" and "source" ones, following *Whiteman et al. (2019)*.

## Trophic position estimation

Trophic position (TP) was determined following a Bayesian approach (see below), employing either bulk- or compound-specific stable isotope analysis methods. To compute the bulk trophic position for all sampled species, we utilized the equation proposed by

*Cabana & Rasmussen (1996)* (Ec.1).

$$TP_\text{bulk} = \lambda + (\delta^{15}N_\text{Consumer} - \delta^{15}N_\text{Baseline})/TEF_\text{bulk}$$

Where $\lambda$ is the trophic position of our baseline organism, $\delta^{15}N_\text{Consumer}$ is the nitrogen isotopic value of each consumer, $\delta^{15}N_\text{Baseline}$ is the nitrogen isotopic value of our baseline, TEF is the nitrogen trophic enrichment factor (TEF) of 3.4‰ (SD =0.51) (*Post, 2002*). Our baseline for this calculation was the $\delta^{15}N$ values of POM as estimated by *Post (2002)*. POM $\delta^{15}N$ values serve as the foundation of the pelagic ecosystem, playing a crucial role in enhancing productivity off the South Orkney Islands which is key for *E. superba,* the main prey of *C. gunnari* (*Everson, 2008*; *Zhu & Zhu, 2022*; *Canseco et al., 2023*). Additionally, to test the accuracy of TP estimates, we calculated *C. gunnari* TP using the equation shown above although the baseline ($\delta^{15}N_\text{Baseline}$) for this calculation was *E. superba,* as it has been mentioned that it is the well-known, main prey.

When estimating trophic position using compound-specific stable isotope analysis, we employed two different methods following the general formula proposed by *Chikaraishi et al. (2009)*:

$$TP_\text{Trc-Phe} = 1 + (\delta^{15}N_\text{Trc} - \delta^{15}N_\text{Phe} - \beta_\text{Trc-Phe})/TEF_\text{Trc-Phe}$$

Where $\delta^{15}N_\text{Trc}$ are the $\delta^{15}N$ values in the trophic AAs glutamic acid (Glx) or proline (Pro), $\delta^{15}N_\text{Phe}$ is the value in phenylalanine (Phe), $\beta_\text{Trc/Phe}$ is the isotopic difference (3.4‰ for Glx and 3.1‰ for Pro) between the corresponding trophic AA and Phe in marine phytoplankton (*Chikaraishi et al., 2009*), and TEFs are empirical trophic enrichment factors between diet and consumer for Glx ($\Delta^{15}N_\text{Glx} - \Delta^{15}N_\text{Phe} = 7.6$ ‰ $\pm 1.2$; *Chikaraishi et al., 2009*) and Pro ($\Delta^{15}N_\text{Pro} - \Delta^{15}N_\text{Phe} = 4.5$ ‰ $\pm 0.9$; *McMahon & McCarthy, 2016*).

Therefore, to estimate TP we implemented Bayesian generalized multivariate multilevel models and statistical routines proposed in this study in the R package *tRophicPosition* v0.8.5 (*Quezada-Romegialli et al., 2018*), using Stan (*Stan Development Team, 2023*), rstan (*Stan Development Team, 2023*) and brms (*Bürkner, 2017*). To parse, compile and fit the models, we used Stan (*Stan Development Team, 2022*), a state-of-the-art platform for statistical modelling and high-performance Bayesian inference with Markov chain Monte Carlo (MCMC) sampling, as implemented in the R packages stan (*Stan Development Team, 2023*) and brms (*Bürkner, 2017*). In both bulk isotope and CSIA analyses, a Gaussian linear model with four chains, each with 2,000 iterations (1,000 burned), was implemented to calculate trophic position with no informative priors. Given that significant correlation was expected within hauls, this grouping factor was treated as a random variable, following a mixed effects approach (*Bürkner, 2017*). Both mean $\delta^{15}N$ baseline and TEF were treated as normally distributed parameters, using the brms function 'me' (*Bürkner, 2017*). These bulk and CSI-AA models were implemented in the R package *tRophicPosition* v 0.8.5 (*Quezada-Romegialli et al., 2018*) in the functions *stanOneBaseline()* and *stanOneBaselineCSIA()*, respectively. This set of functions propagate the error term of $\beta$ and TEF. The code can be accessed at https://github.com/clquezada/tRophicPosition/tree/stanCSIA.

The three trophic position estimation approaches were compared within and across consumers using medians, credibility intervals and the probability of differences derived from the posterior distributions from each model fit (*Kruschke, 2010*).

## RESULTS

### Bulk and individual amino acid δ15N by species

Among the different species analyzed, particulate organic matter (POM) exhibited the lowest mean $\delta^{15}N_{bulk}$ value of $-0.5‰$ ($\pm1.4$), followed by *E. superba,* whose mean $\delta^{15}N_{bulk}$ was 3.5‰ ($\pm0.6$). The highest $\delta^{15}N_{bulk}$ values were observed for *C. gunnari* and *E. antarctica,* with mean δ15Nbulk values of 8.5‰ ($\pm0.2$) and 8.6‰ ($\pm0.7$), respectively (Table 1, Fig. 1). Regarding individual amino acid, nitrogen isotopic values were highly variable among species which ranged from more than 23‰ in glutamic acid (Glx) to -22‰ in threonine (Table 1, Fig. 1). Values of $\delta^{15}N$ close to zero were observed in glycine, and serine, although differences of >3‰ were observed between species. Negative values were observed among all species for glycine and threonine. Overall, *E. superba* had lower values than the other species in all amino acids, except phenylalanine. *Champsocephalus gunnari* and *Electrona antarctica* had higher values for all amino acids than invertebrates (Table 1, Fig. 1), especially isoleucine, leucine, lysine, and valine (>2‰). *Antarctomysis maxima* and *Themisto gaudichaudii* had higher values than *Euphausia superba* and *Euphausia frigida.*

Electrona antarctica displayed the largest difference in Phe and Glx (ΔGlx-Phe) with a mean value of 23.1‰ ($\pm1.0$) followed by *C. gunnari* with 21.8‰ ($\pm0.7$) while the lowest difference was observed for *E. superba* ($13.2 \pm 2.2‰$, Fig. 1). *Euphausia superba* also displayed the lowest difference between Phe and Pro (ΔPro-Phe) with a mean value of 9.7 ($\pm2.2‰$).*Themisto gaudichaudii, E. frigida* and *A. maxima* showed smaller differences ($\sim1‰$) when using ΔPro-Phe than when using ΔGlx-Phe. *Champsocephalus gunnari* and *E. antarctica* values were slightly smaller when ΔPro-Phe ($\sim20‰$) was used instead of ΔGlx-Phe ($\sim22‰$).

### Trophic position

Estimated TP varied depending on the specific method employed (Table 2, Fig. 2). TPs computed using bulk-SIA were similar to those computed using the Glx-Phe CSI-AA approach but showed more variability. Median $TP_{Glx-Phe}$ estimates resulted in 0.01 levels higher, $p(TP_{Glx-Phe} \geq TP_{bulk}) = 0.54$ (*i.e.,* probability of $TP_{Glx-Phe}$ being higher than $TP_{bulk}$), than median bulk-SIA estimates, across all consumers which indicates that there was no difference between methods. $TP_{Pro-Phe}$ estimates ranged between 0.2 and 0.8 levels lower than $TP_{bulk}$ estimates, depending on the consumer, with a $p(TP_{Pro-Phe}<TP_{Bulk}) = 0.92$ across all consumers. Similar differences were observed for *C. gunnari* TP estimates, whose median values reached 3.6, 3.4 and 3.1 (Table 2), using bulk-SIA, Glx-Phe and Pro-Phe CSI-AA , respectively.

Overall, there was greater disagreement between methods for invertebrates than for fish, although fish were consistently 1 TP higher than invertebrates. *E. antarctica* had a trophic level of 3.6 using bulk SIA, 3.5 using Glx-Phe and a TP of 3.3 when using Pro-Phe (Table 2). Unexpectedly, *A. maxima* appeared at a higher trophic position than the rest of invertebrates when using Glx-Phe (TP =3.2; Table 2) and its estimated TP values were approximately 0.4 higher than the rest of invertebrates. The lowest TP (1.9–2.2) was observed for *E. superba* (Table 2), while the trophic positions of *T. gaudichaudii* (2.1–2.6) and *E. frigida* (2.0–2-8) were similar regardless of the method (Table 2, Fig. 2).

**Table 1  $\delta^{15}$N of bulk and individual amino acid by species.** Mean ($\pm$SD) $\delta^{15}$N of bulk and individual amino acid by species sampled during summer 2019 off the South Orkney Islands onboard RV Kronprins Haakon and FV Antarctic Endeavor. Muscle tissue was sampled for *C. gunnari* and *E. antarctica* whereas pools of five individuals were performed for the remaining species.

| | Amino acid | POM | *Euphausia superba* | *Themisto gaudichaudii* | *Euphausia frigida* | *Antarctomysis maxima* | *Electrona antarctica* | *Champsocephalus gunnari* |
|---|---|---|---|---|---|---|---|---|
| Bulk | $\delta^{15}$N$_{bulk}$ | $-0.5 \pm 1.4$ | $3.5 \pm 0.6$ | $4.9 \pm 0.8$ | $5.4 \pm 0.3$ | $5.9 \pm 0.4$ | $8.6 \pm 0.7$ | $8.5 \pm 0.2$ |
| Trophic | Alanine | – | $13.7 \pm 2.5$ | $21.1 \pm 1.9$ | $17.8 \pm 0.8$ | $21.0 \pm 1.2$ | $23.2 \pm 1.4$ | $22.5 \pm 0.8$ |
| | Aspartic acid | – | $11.4 \pm 1.5$ | $14.7 \pm 0.7$ | $14.4 \pm 0.5$ | $15.7 \pm 0.6$ | $17.9 \pm 0.9$ | $16.9 \pm 0.5$ |
| | Glutamic acid | – | $14.9 \pm 1.8$ | $19.4 \pm 0.4$ | $18.5 \pm 0.2$ | $21.8 \pm 0.6$ | $23.9 \pm 1.6$ | $23.1 \pm 0.4$ |
| | Glycine | – | $-3.5 \pm 1.1$ | $1.7 \pm 0.9$ | $-2.1 \pm 0.9$ | $-0.4 \pm 0.9$ | $-1.1 \pm 1.4$ | $-1.7 \pm 1.0$ |
| | Isoleucine | – | $10.2 \pm 1.9$ | $17.9 \pm 1.4$ | $9.8 \pm 1.6$ | $16.6 \pm 1.1$ | $22.0 \pm 1.5$ | $22.2 \pm 1.7$ |
| | Leucine | – | $10.8 \pm 1.1$ | $17.1 \pm 1.1$ | $11.9 \pm 0.6$ | $18.0 \pm 0.5$ | $21.0 \pm 1.3$ | $21.4 \pm 0.6$ |
| | Proline | – | $11.4 \pm 0.9$ | $14.4 \pm 0.6$ | $12.7 \pm 1.4$ | $15.5 \pm 0.7$ | $21.5 \pm 1.3$ | $19.8 \pm 0.6$ |
| | Valine | – | $12.9 \pm 1.6$ | $18.8 \pm 0.6$ | $14.1 \pm 0.2$ | $18.2 \pm 0.9$ | $22.2 \pm 1.4$ | $21.5 \pm 0.9$ |
| Source | Lysine | – | $2.1 \pm 0.8$ | $1.2 \pm 1.5$ | $2.9 \pm 0.4$ | $0.8 \pm 1.0$ | $0.2 \pm 2.0$ | $2.5 \pm 1.3$ |
| | Methionine | – | $-1.3 \pm 1.7$ | $3.7 \pm 0.6$ | $0.7 \pm 0.9$ | $4.2 \pm 1.0$ | $2.8 \pm 2.6$ | $1.4 \pm 0.9$ |
| | Phenylalanine | – | $1.7 \pm 1.0$ | $2.6 \pm 1.0$ | $1.5 \pm 1.3$ | $1.8 \pm 0.6$ | $0.8 \pm 0.7$ | $1.3 \pm 0.8$ |
| | Tyrosine | – | $5.0 \pm 1.4$ | $5.1 \pm 0.9$ | $6.4 \pm 1.0$ | $5.1 \pm 0.9$ | $6.1 \pm 1.3$ | $6.3 \pm 1.3$ |
| Other | Serine | – | $-0.9 \pm 2.7$ | $3.7 \pm 1.3$ | $2.3 \pm 0.5$ | $3.7 \pm 1.3$ | $2.7 \pm 1.1$ | $-3.9 \pm 0.9$ |
| | Threonine | – | $-11.9 \pm 1.4$ | $-17.9 \pm 2.2$ | $-11.0 \pm 1.0$ | $-17.9 \pm 2.2$ | $-15.9 \pm 0.7$ | $-19 \pm 0.8$ |

## Assessing relative accuracy in *C. gunnari* TP

The use of *E. superba* as baseline when estimating TP using bulk-SIA (TP$_{bulk}$) allowed us to assess precision of our TP estimates. The $\Delta$Glx-Phe method overestimated the difference between *C. gunnari* and *E. superba* TP by 0.1 (CI$_{95\%}$: 0.0$-$0.4) TP compared to the theoretical difference of 1 TP. The use of $\Delta$Pro-Phe yielded a 0.2 (CI$_{95\%}$: 0.1$-$0.4) difference between the expected difference and the one calculated, while the use of bulk SIA overestimated *C. gunnari* TP by 0.4 (CI$_{95\%}$: 0.1$-$1.0) (Fig. 3).

## DISCUSSION

In this study we estimated the TP of *C. gunnari* and its main prey using bulk and CSI-AA around the South Orkney Islands. To do so, we applied and compared different analytical approaches in a common Bayesian framework using routines recently implemented in the tRophicPosition R package v0.8.5 (*Quezada-Romegialli et al., 2018*). This approach allows researchers to incorporate uncertainty regarding important parameters like $\beta$ and TEF, parameters known to be highly variable within and between species, diet quality, taxa and habitat (*McMahon et al., 2015*; *McMahon & McCarthy, 2016*; *Ramirez et al., 2021*). Our results indicate that the trophic position of *C. gunnari* is between 3.1 and 3.6; hence, our estimates are method-dependent. The use of bulk and compound-specific stable isotope analysis of amino acids is valuable, allowing us to better understand the complex trophic web around *C. gunnari*. This analytical approach aids in the species recovery by facilitating the development of appropriate management strategies focused on protecting all the organisms in its trophic network.

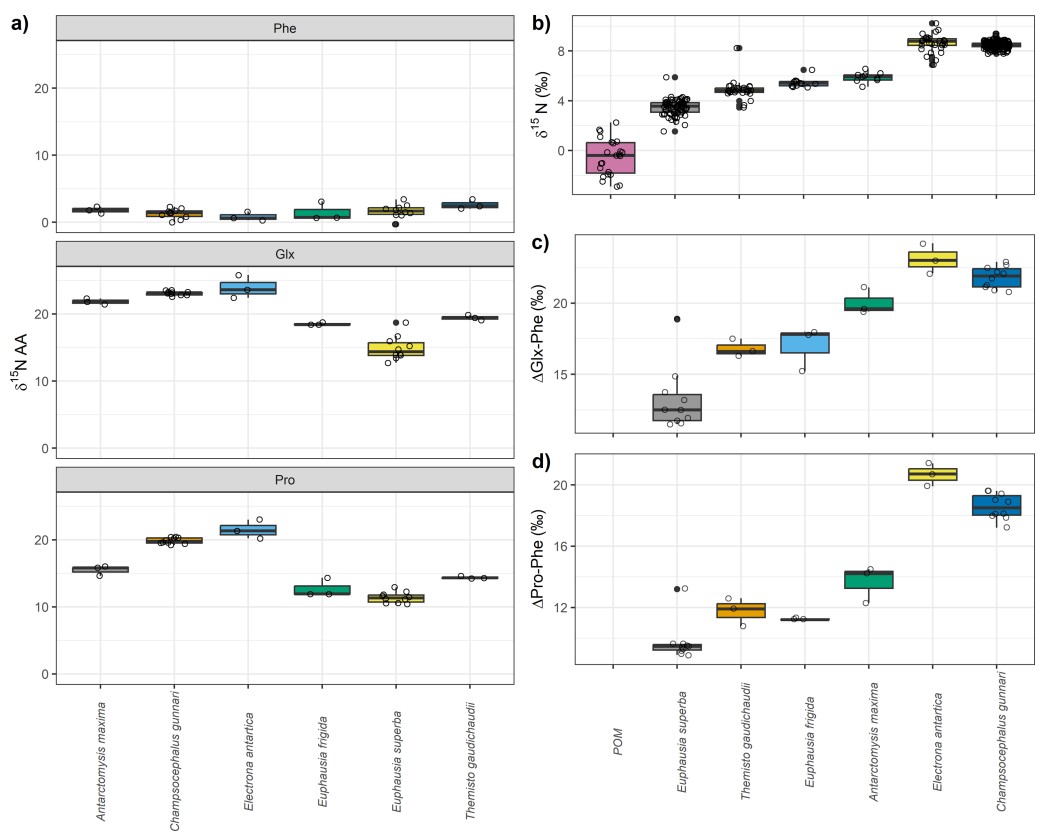

**Figure 1** **Nitrogen compound specific stable isotopes of amino acids, bulk $\delta^{15}$N values and differences between glutamic acid (Glx) and phenilananine (Phe) and proline (Pro) and phenylalanine (Phe).**
(A) Nitrogen compound specific stable isotopes of amino acids per species. Trophic and source nitrogen amino acids are shown following *Whiteman et al.*'s (*2019*) classification and the estimated mean value for source and trophic. Variation in (B) bulk $\delta^{15}$N values, (C) differences between glutamic acid (Glx) and phenilananine (Phe) and (D) between proline (Pro) and phenylalanine (Phe).

**Table 2** **Calculated trophic position per species.** Calculated trophic position for each species considering bulk stable isotope analysis (SIA, $TP_{bulk}$), the difference in $\delta^{15}$N between amino acids Glx and Phe ($TP_{Glx\text{-}Phe}$) and the difference in $\delta^{15}$N between amino acids Pro and Phe ($TP_{Pro\text{-}Phe}$) which make use of compound specific stable isotope analysis in amino acids (CSI-AA).

| Species | Median trophic position (95% Credibility intervals) | | |
|---|---|---|---|
| | $TP_{bulk}$ | $TP_{Glx-Phe}$ | $TP_{Pro-Phe}$ |
| *Euphausia superba* | 2.2 (1.9–2.8) | 2.3 (2.1–2.5) | 1.9 (1.8–2.0) |
| *Themisto gaudichaudii* | 2.6 (2.2–3.3) | 2.7 (2.5–3.0) | 2.2 (2.0–2.4) |
| *Euphausia frigida* | 2.8 (2.4–3.6) | 2.8 (2.5–3.0) | 2.0 (1.9–2.3) |
| *Antarctomysis maxima* | 2.8 (2.4–3.7) | 3.2 (2.9–3-5) | 2.4 (2.1–2.6) |
| *Electrona antarctica* | 3.6 (3.0–4.7) | 3.6 (3.3–3.9) | 3.4 (3.1 -3.6) |
| *Champsocephalus gunnari* | 3.6 (3.0–4.8) | 3.4 (3.2–3.6) | 3.1 (3.0–3.3) |

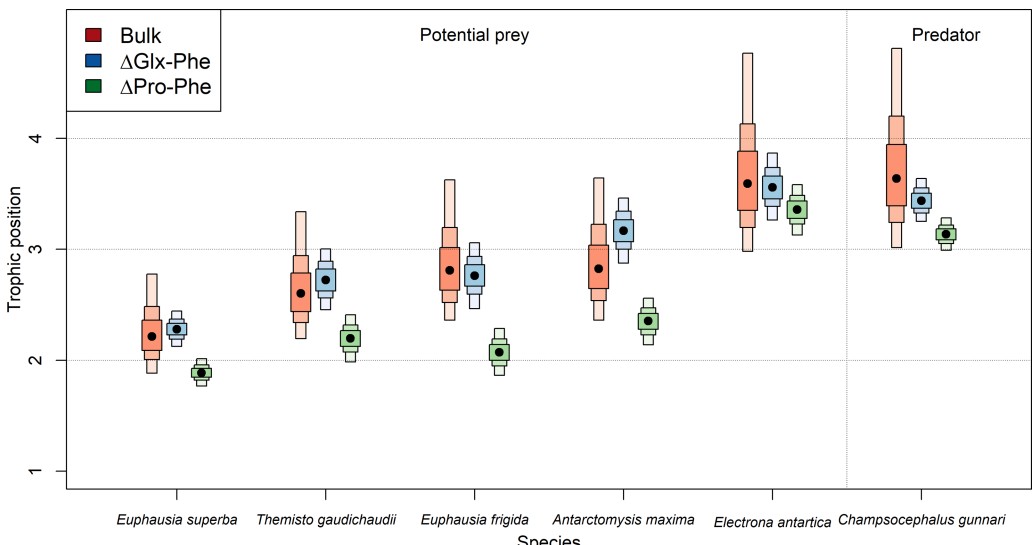

**Figure 2** **Predicted median estimated trophic positions.** Predicted median (dot) and 95% credibility intervals of the estimated trophic position (TP) by species (i) using bulk SIA and the equation proposed by *Cabana & Rasmussen (1996)*, (ii) using ΔGlx-Phe as proposed by *Chikaraishi et al. (2009)* and (iii) ΔPro-Phe as proposed by *McMahon et al. (2016)*.

Asumming that the diet of *C. gunnari* primarily consists of *E. superba*, an expected difference in TP of approximately 1 between the two species was anticipated. However, discrepancies arise when applying the TEF proposed by *Post (2002)*, leading to a biased and more variable TP in *C. gunnari*. This deviation from the observed trophic interactions between both species (*Main et al., 2009*; *Zhu & Zhu, 2022*; *Canseco et al., 2023*) highlights the importance of the TEF when estimating the TP. The use of a fixed $TEF_{bulk}$ of 3.4 (*Post, 2002*) may not be realistic as suggested by *Canseco, Niklitschek & Harrod (2022)*, who propose that $TEF_{bulk}$ is influenced by factors such as temperature, prey baseline, and prey-type, especially in fish species. Adjusting the TEF for temperature of 5.2 (±0.68)‰ was applied, as recommended by *Canseco, Niklitschek & Harrod (2022)* for fish, resulted in a TP difference of 0.93 between the species, in better agreement with the expectations compared to the 1.4 difference observed with *Post (2002)* TEF.

In terms of TP estimation using CSI-AA, our findings indicate that median TP of *C. gunnari* was 0.5 higher when using the ΔGlx-Phe ($TP_{Glx-Phe}$; *Chikaraishi et al., 2009*) than when using the ΔPro-Phe ($TP_{Pro-Phe}$; *Ramirez et al., 2021*). This discrepancy could be attributed to Pro being derived from Glx and having a longer turnover rate due to its lesser involvement in transamination reactions driving Glx $\delta^{15}N$ (*Goto et al., 2018*). It is essential to acknowledge the limitations due to the small size of our sample and that all TP methods are susceptible to uncertainty due to TEF assumptions. Thus, if a lower $TEF_{Glx-Phe}$ of 5.5‰ were used for fishes, as *McMahon et al. (2015)* suggested for middle and upper-trophic-level species, a much higher trophic position (TP =4.3; $CI_{95\%}$: 4.0–4.7) was estimated. The use of other trophic AA's, based on a multi-AA approach, to estimate TP, also proposed by several researchers (*Chikaraishi et al., 2009*; *Nielsen, Popp & Winder,*

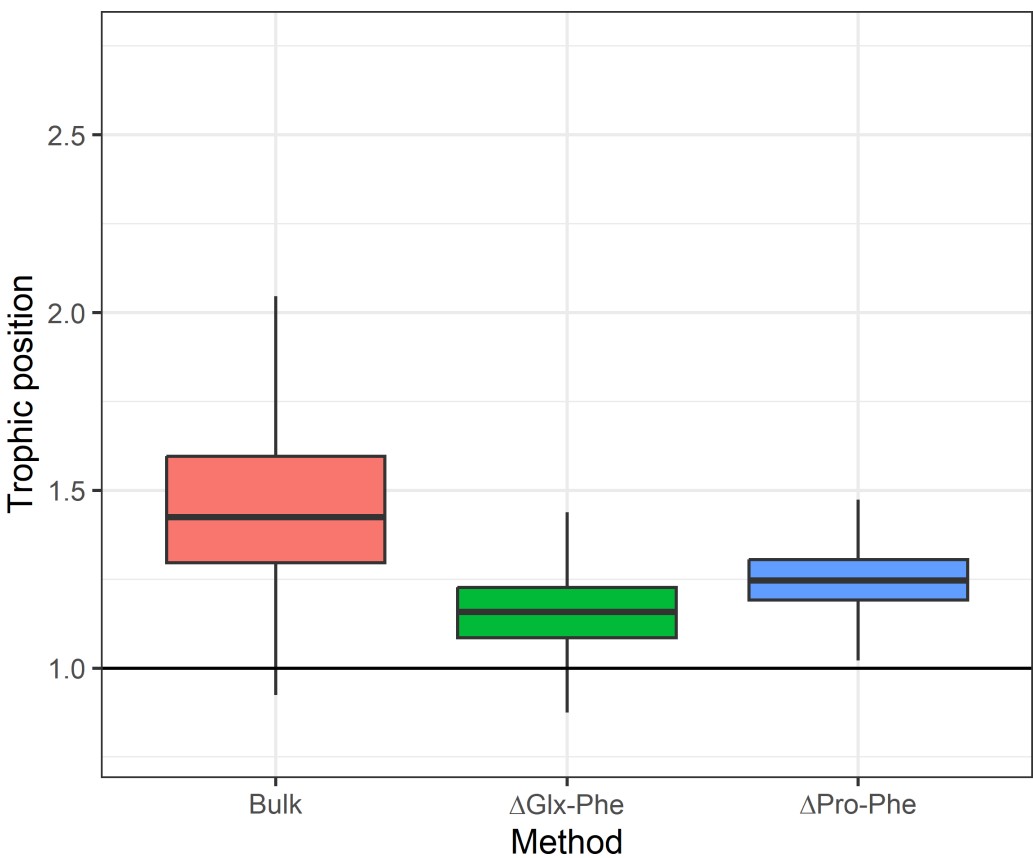

**Figure 3** **Expected difference in trophic position between *Euphausia superba* and *Champsocephalus gunnari*.** Expected difference in trophic position ($\Delta$TP =1) between *Euphausia superba* and *Champsocephalus gunnari* and the observed difference obtained by applying the three tested methods. Bulk TP results were obtained by using *E. superba* as our baseline organism.

*2015*; *McMahon & McCarthy, 2016*), overestimated the TP of *C. gunnari,* yielding TPs 1.0 using the $\Delta$Ala-Phe (TP$_{\text{Ala}-\text{Phe}}$ =4.3; CI$_{95\%}$: 3.8–4.8) and 1.9 using $\Delta$Ile-Phe (TP$_{\text{Ile}-\text{Phe}}$ =5.2; CI$_{95\%}$: 4.6–6.0). Such results support the use of either $\Delta$Glx-Phe or $\Delta$Pro-Phe as they provide values closer to the expected value.

TEF sensitivity was also noted by *Bradley et al. (2015)* where they observed that 80% of the variability of TP estimates using SCA and CSI-AA was explained by the TEF. Additionally, they derived a Glx-Phe TEF of 5.7 $\pm$ 0.3 which deviates by 2‰ from the value suggested by *Chikaraishi et al. (2009),* a value utilized in our current study. This difference in methodologies and results highlights the uncertainty and sensitivity of TEF, even when using CSI-AA. We observed that, as in many bulk SIA applications (*Post, 2002*; *Canseco, Niklitschek & Harrod, 2022*), TEF remains the most sensitive parameter. Consequently, more research should be focused given its potential to under or over estimate TP across various organisms and ecosystems (*Germain et al., 2013*; *Bradley et al., 2015*; *Choy et al., 2015*; *Lorrain et al., 2015*). The sensitivity in TEF could be attributed to different physiological processes affecting $\delta^{15}$N from different metabolic pathways,

potentially resulting in enrichment (*Ohkouchi et al., 2017*; *Goto et al., 2018*; *Ishikawa, 2018*).

Using POM $\delta^{15}N_{bulk}$ values as baselines to estimate the trophic position using bulk stable isotope data for all species may explain the large variability observed compared to CSI-AA methods. Nitrogen isotopic baselines are mostly driven by nutrient availability and utilization (*Brault et al., 2018*; *Brault et al., 2019*; *Ramirez et al., 2021*), and POM $\delta^{15}N_{bulk}$ values are known to vary spatially with higher $\delta^{15}N$ linked with coastal, productive waters and smaller $\delta^{15}N$ linked to offshore waters and within small time-frames in the Southern Ocean (*Brault et al., 2019*; *St John Glew et al., 2021*). Both fish species, *C. gunnari* and *E. antarctica*, had nearly the same trophic position ($\sim$3). Despite being smaller and sometimes consumed by *C. gunnari* (*Canseco et al., 2023*), *E. antartica* had even a slightly higher TP when applying $\Delta$Pro-Phe and $\Delta$Glx-Phe estimation methods. This similarity in trophic positions between *E. antarctica* and *C. gunnari* might be explained if both species had different nitrogen isotopic baselines (*St John Glew et al., 2021*) as found in Weddell and Ross seals (*Brault et al., 2018*; *Brault et al., 2019*; *Ramirez et al., 2021*). Available evidence indicates that both species have different spatial and bathymetric distributions, different foraging strategies and diet (*Kock et al., 2012*; *Saunders et al., 2014*).

CSI-AA had been previously measured in *Euphausia superba* collected from different parts of the Southern Ocean (*Schmidt et al., 2004*; *Schmidt et al., 2006*). While our results were not separated by sex, they were similar to the data of *Schmidt et al. (2006)* for males. However, we found large differences (>2‰) between our and their results regarding females, at least for specific amino acids, such as aspartic acid. These differences might be explained by sex-related effects on the physiology, feeding habits, prey selection, foraging strata used by reproductive females, and year- or site-specific variability (*Schmidt et al., 2006*; *Atkinson et al., 2008*).

This study highlights the sensitivity and variability of using different methods to estimate the trophic position of any group of species. Each tested method relies on a set of assumptions that may be influenced by the ecophysiology of individual species, hence, to certain physiological characteristics such as *e.g.*, the different metabolic pathways driving the isotopic composition of individual amino acids in fish. Moreover, more uncertainty around our median trophic position estimates for each species was observed when using bulk or the $\Delta$Glx-Phe method with CSI-AA than with $\Delta$Pro-Phe method. In addition, the $\Delta$Pro-Phe method yielded TP estimates that seem more realistic with the observed diet of *C. gunnari*, so it may be more suitable for species inhabiting harsh environments like the Southern Ocean. In conclusion, the use of CSI-AA to estimate TP yields relatively similar but clearly more precise results than bulk estimates.

### Funding

Jose Canseco was supported by a doctoral fellowship from Universidad de Los Lagos (Chile). Edwin Niklitschek was supported by the Chilean Antarctic Institute (INACH)

through grant RT_68-18 and by Universidad de Los Lagos through Research Grant RT 29/18. Chris Harrod and Claudio Quezada-Romegialli were supported by Núcleo Milenio INVASAL funded by Chile's ANID—Programa Iniciativa Científica Milenio (NCN16_034 and NCN2021_056). The funders had no role in study design, data collection and analysis, decision to publish, or preparation of the manuscript.

## Grant Disclosures

The following grant information was disclosed by the authors:

Universidad de Los Lagos (Chile).

Chilean Antarctic Institute (INACH).

Núcleo Milenio INVASAL.

## Competing Interests

The authors declare there are no competing interests.

## Author Contributions

- Jose Antonio Canseco conceived and designed the experiments, performed the experiments, analyzed the data, prepared figures and/or tables, authored or reviewed drafts of the article, and approved the final draft.
- Edwin J. Niklitschek conceived and designed the experiments, analyzed the data, prepared figures and/or tables, authored or reviewed drafts of the article, and approved the final draft.
- Claudio Quezada-Romegialli analyzed the data, authored or reviewed drafts of the article, and approved the final draft.
- Chris Yarnes performed the experiments, analyzed the data, authored or reviewed drafts of the article, and approved the final draft.
- Chris Harrod conceived and designed the experiments, analyzed the data, prepared figures and/or tables, authored or reviewed drafts of the article, and approved the final draft.

## Field Study Permissions

The following information was supplied relating to field study approvals (*i.e.*, approving body and any reference numbers):

Field experiments were approved by the Instituto Antartico Chileno (RT_68-18) and a permit is provided by the Office of the Commissioner Government House.

## Data Availability

The raw measurements are available in the Supplementary Files.

## Supplemental Information

Supplemental information for this article can be found online at http://dx.doi.org/10.7717/peerj.17372#supplemental-information.

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
