# Peer review of "Comparing trophic position estimates using bulk and compound specific stable isotope analyses: applying new approaches to mackerel icefish Champsocephalus gunnari"

_PeerJ, doi:10.7717/peerj.17372_

## Round 0.1 · original submission · Minor Revisions

The reviewers appreciate the clarity and structure of your manuscript. They suggest elaborating on the confounding factors, particularly the Trophic Enrichment Factor (TEF), for a more comprehensive understanding. They also recommend adding references by Ishikawa (2018) and Bradley et al. (2015) to enrich your literature review.

The originality and sound methods of your research were acknowledged. However, the first reviewer expressed interest in seeing multi-AA approaches for TP-estimation explored in your work. Lastly, they found your data robust and your conclusions well-stated, but suggested a comparison to similar studies, such as Bradley et al. (2015), for a broader perspective.

Please consider these and the detailed line-wise points in your revisions.

·

Basic reporting

The Manuscript „Comparing trophic position estimates using bulk and compound specific stable isotope analyses: applying new approaches to mackerel icefish Champsocephalus gunnari” by Rodriguez et al. cleverly utilizes the strict dependency in their predator-prey system to evaluate different technics based on stable nitrogen isotope analyses for trophic position estimation. The manuscript is largely well written. I found a few specific paragraphs complicated to read, which I address in detail in the specific comments below.

The authors use most of the state-of-the-art literature available to the topic. On review, which I found quite interesting and that is missing from the author’s list might be: Ishikawa, N. F. (2018). Use of compound-specific nitrogen isotope analysis of amino acids in trophic ecology: assumptions, applications, and implications. Ecological research, 33(5), 825-837.

Another large study evaluation CSIA-AA for TP-estimates (though vs. gut content analysis) is: Bradley, C. J., Wallsgrove, N. J., Choy, C. A., Drazen, J. C., Hetherington, E. D., Hoen, D. K., & Popp, B. N. (2015). Trophic position estimates of marine teleosts using amino acid compound specific isotopic analysis. Limnology and oceanography: Methods, 13(9), 476-493. Interestingly, they found a deviation required for their TEF-factor estimates to fit the data properly, which might be also of interest for the Author’s to compare to their analysis.

The article, figures and table have a professional structure. As mentioned specifically below, I recommend the combination of two figures due to redundance with tables.
The results are relevant for addressing the hypothesis postulated by the authors.

Experimental design

I did not find anything contradicting the Aims and Scope of the Journal. I would define the research itself as a “method evaluation”, with large ecological implications. TP-estimates are frequently used in ecology for food web reconstruction and monitor of structural changes. For that purpose the analysis used by the authors are appropriate and state-of-the-art. A few minor information were missing regarding the method description, as addressed in the specific comments below.

One optional thing, that I would personally be interested in is: Larsen at al. (2015, cited by the authors) also suggested multi-AA approaches for TP-estimation. Though they did not find any major advances in such an approach, the authors do have the required data, and claim to perform a thourough method evaluation. I was thus wondering, if they would also perform TP-estimation using such other approaches that can be found in the literature.

Validity of the findings

The data provided is robust and statistically sound and controlled. The conclusions are well stated. One thing that I would like to see as well is a comparison to similar comparative studies, such as Bradley, C. J., Wallsgrove, N. J., Choy, C. A., Drazen, J. C., Hetherington, E. D., Hoen, D. K., & Popp, B. N. (2015). Trophic position estimates of marine teleosts using amino acid compound specific isotopic analysis. Limnology and oceanography: Methods, 13(9), 476-493. Changes in the TEF-parameter estimates might be very important, especially variation across different systems. Thus I think at least mentioning and linking these papers would make it easier for readers specifically looking for such estimates to find the appropriate one for their system.

Additional comments

Specific comments:
L48: Recommend that you specify „bulk stable isotope analysis”
L58: This is not 100% true. CSSIA might still be affected, e.g. by fractionation during assimilation (i.e., trophic enrichment factors are still required) and by catabolic processes, e.g., if the essential component is educt in an biochemical pathway the remaining pool might get enriched compared to its source.
L69: Maybe mention that these estimates vary quite a lot within literature (I have seen 2.7 to 3.8) which has quite an impact, especially at higher TPs.
L70: Space too much
L71: Space missing
L104-L107: This part is already a bit too specific for the introduction and I would recommend moving it to the methods. At this point, I would rather emphasize what benefit regarding your hypothesis and/or the knowledge about C. gunnari you hope to get by using a Bayesian approach.
L121: That is unfortunate, it would be great to have also the comparison vs. the baseline for all methods. Would it be possible to still analyze some of the samples?
L139: Did you also modify the C-terminus? It would be great if you can cite a reference for your derivatization method for more details. Otherwise, please explain the reagents and methods used in the preanalytical procedure in more detail.
L175: Please provide a rational/reference for using this specific TEF.
L186: I think you can simplify this part by using a variable for Glx/Pro in the equation and then afterwards explaining that this variable stands for either amino acid.
L207: The link did not work for me. Please check again.
L217: Please mention Figure 1 before Figure 2. I was also wondering, if it requires both figures, or whether they might be combined into 1. Why showing all AA in the plot and additionally list their values in a table? Maybe just select those you are using in your models for the plot and list the rest in the table.
L227-L234: This paragraph needs rephrasing. E.g. “The difference… indicated that E.superba had the lowest value” does not really make sense. Furthermore, I would recommend e.g., “E.antarctica displayed the largest differences between Phe and Glx…”, instead of first “defining the difference” and then stating that “X had the highest value.
L239: I know that there is now significance in Bayesian statistics, but still the “effect” of Delta-TP of 0.01 seems negligible. The difference to method 3 (Pro-Phe) is however quite a lot. What I was wondering is, if the TP-estimates for Pro-Phe correlated with those of Glx-Phe, which would indicate that an adjustment for the TEF for Pro-Phe could produce comparable results.
L258: 0.1 TP is pretty good, considering that a lot of the factors are pure estimates including some standard error. This also makes me wonder, whether the R package you used for TP estimation also propagates the error of said factors, and whether you have considered that for your analysis. Furthermore, could you provide 95% CI for these TP estimates?
L297: You mention that already 4 lines above, consider deleting this sentence.
L283-L300: The structure of this pararaph is a bit confusing for the reader. I would recommend, that you start with the expected value and why (Delta-TP=1) and which method came closest. Then start listing reasons of why the methods might deviate from the expected value and which other parameter-assumptions provided better results.
L316: I would skip the Conclusion, since you have such a short Discussion anyway. At least, do not bring up new ideas in the Conclusion. L324 is such a new idea, which is great, and one that I had already earlier on, but please move this idea already to an appropriate location within the discussion. Another citation supporting this idea is: Goto, A. S., Miura, K., Korenaga, T., Hasegawa, T., Ohkouchi, N., & Chikaraishi, Y. (2018). Fractionation of stable nitrogen isotopes (15N/14N) during enzymatic deamination of glutamic acid: Implications for mass and energy transfers in the biosphere. Geochemical Journal, 52(3), 273-280.

Reviewer 2 ·

Basic reporting

The manuscript titled "Comparing trophic position estimates using bulk and compound specific stable isotope analyses: applying new approaches to mackerel icefish Champsocephalus gunnari" examines the trophic position of two fish species and four invertebrate species in the Southern Ocean using three different calculations and assessing their differences. The MS uses clear and unambiguous language, is well-structured, and mainly easy to follow. Mostly, sufficient background is given in the MS to understand the topic, however, I think more information regarding the possible confounding factors of the calculations, especially TEF should be included throughout. Figures and tables support the text, and the results are relevant to the posed hypothesis.

Abstract
Line 22: What are realistic field conditions? Maybe specify more or leave it out of the abstract and only explain it in the text.
Line 24: Amino acid --> two words
Line 27: Unclear why samples were taken from four other invertebrate and fish species --> missing here what is mentioned in the Introduction that those are also possible prey species of C. gunnari
Line 32: The second TP seems superfluous in this sentence

Introduction
Line 43: as well as --> 2nd as missing.
Line 64: represents --> s at the end missing
Line 66: Do you mean here “in respect to” or “respective to”?
Line 70: The rationale -->The missing at the start of the sentence
Line 71: Researchers discriminate -->two words
Line 71: Full stop before the citation bracket
Line 90: Maybe a map of the Southern Ocean would be nice here to show area 48, or at least explain to readers where area 48 is roughly located.
Lines 104-107: I am missing in this sentence what you used the models for.

Figures & Tables
Table 2: I would change the title of this table to “Calculated trophic position…” rather than “Predicted trophic position” --> predicted sounds more based on feeding observations rather than using SI methods.

Experimental design

The MS is original, primary research with a well-defined hypothesis that fills a knowledge gap posed in the text. The methods are described sufficiently for replication and sound for the approach. The code is made available and packages used for analysis are mentioned thoroughly.

Methods
Line 112: Individual C. gunnari --> s too much
Line 113: What do you mean by “pooled” samples?
Line 115: Does the n=50 mean for each invertebrate prey species (total 150 samples) or does it mean a total of 50 samples?
Line 126: How long were samples oven-dried?
Line 134: 0.11 --> space between number and permille sign is missing
Line 172: TPbulk --> b needs to be a subscript, too.
Line 180: shown above instead of showed
Line 197: The acronym explanation for MCMC is missing.

Validity of the findings

All underlying data is provided, is statistically sound, and conclusions are linked to the original hypothesis, However, I would have liked to see a mention of the biggest factor, the uncertainty in TEF use.

Results
Line 227: In this paragraph, it is not explicitly mentioned that you are talking about mean values. It is implied because of the previous paragraph but should maybe be stated here for clarity.
Line 231: I don’t quite understand the word of the use “however” here because it expresses a contradiction to what has been said before, but I do not see a contradiction of results mentioned in this sentence to the previous sentences.
Line 239: resulted in --> in missing here? In general, this sentence is hard to understand.
Line 241: Did you test for no difference between methods statistically?
Line 246: If you want to remind the reader of the TEFs that you used for each calculation, I would do it in the above paragraph when you start talking about TP results rather than in this paragraph. It makes it seem like you used different TEFs in the above paragraph and forgot to mention them there.
Line 250: I would stick to just using the acronym TP rather than a mix of the acronym and the words (also in other places throughout the MS like in line 293). Also, A maxima appeared higher than what? Does this also refer to the other invertebrates like the end of the sentence? Or do you mean higher than what you expected?

Discussion
Line 263: I don’t think there is a need to re-introduce your acronym TP, CSIA-AA or to spell out the full species name again.
Line 274: How would knowing the exact trophic position aid in the recovery of the species?
Lines 275-282: This paragraph seems a bit out of place and interrupts the C. gunnari story.
Line 286: I find susceptible the wrong word here unless you add something along the lines of “susceptible to uncertainty due to TEF assumptions”.
Line 292: the word “is” is superfluous here
Line 293: replaces “to be” with “is”
Lines 295-296: Is that difference statistically significant or does it lead to a substantial difference in terms of diet or ecological meaning?
Lines 283-300: Aren’t you showing in this paragraph though that choosing a different TEF leads to more substantial differences in TP than using a different calculation method? Hence, shouldn’t we be way more skeptical about which TEF to use than which calculation method?
Line 306: “smaller d15N linked to” --> linked missing. And “within small time-frames” --> a superfluous.
Line 308: Do you mean trophic level here or trophic position? If you mean trophic level, it may be worth explaining to the reader what the difference is.
I would have liked to see more of a discussion around the TEF values as they made a bigger difference than bulk versus CSIA.

---

## Round 0.2 · accepted · Accept

All reviewer comments have been addressed in a sufficient and explanatory way and considering the reviewer suggestions, I believe your manuscript is ready for publication. Congratulations to your nice work and thanks to the reviewers for their careful evaluation!

·

Basic reporting

I think the authors did an excellent job in revising the manuscript. I have no further comments and would recommend the acceptance.

Experimental design

no comment

Validity of the findings

no comment

Additional comments

no comment